## [Reviewer comments · BMJ Open]

ARTICLE DETAILS

TITLE (PROVISIONAL)	Development of the DONOR prediction model on the risk of hypertensive complications in oocyte donation pregnancy: study protocol for a multicenter cohort study in The Netherlands
AUTHORS	Lafeber, Géraldine; Van der Endt, Vera H.W.; Louwers, Yvonne; le Cessie, S; van der Hoorn, Marie-Louise; Lashley, Eileen

VERSION 1 – REVIEW

REVIEWER	Simeone, Serena University Hospital Careggi
REVIEW RETURNED	24-Sep-2023

GENERAL COMMENTS	The paper presents a study protocol which aims to address the possible limitations in current models for predicting PIH in donor oocyte pregnancies. The study is well designed but incomplete, as, in my opinion: - HLA-mismatch tests fall into genetic testing and therefore its costs are higher than current prediction models. The Authors should at least mention how countries/centers could afford these tests on a large population;- The analysis should include a comparative evaluation of the traditional tests on the same population, otherwise no significant difference could be calculated, since, as the Authors themselves admit, no data is published on the predictive performance of the available screenings on donor oocyte population. Last but not least, I suggest the use of a respectful language in the Introduction section: - some sentences like the second one, lines 66-67 and 74-75, could be amended avoiding old fashioned expressions like "produce viable oocytes for healthy offspring" (not acceptable in 2023) and "lesbian motherhood".
---

VERSION 1 – AUTHOR RESPONSE

Reviewer 1 (Serena Simeone)'s comments (08-02-2024):

*The paper presents a study protocol which aims to address the possible limitations in current models for predicting PIH in donor oocyte pregnancies.

The study is well designed but incomplete, as, in my opinion:

HLA-mismatch tests fall into genetic testing and therefore its costs are higher than current prediction models. The Authors should at least mention how countries/centers could afford these tests on a large population;

- We acknowledge the importance of considering affordability of such tests, particularly on a large scale within different centers. However, it should be noted that the predictor ‘fetal-maternal HLA-(mis)matches’ will not be used as a candidate predictor in the model yet. This is primarily because prenatal HLA-typing cannot yet be performed, hence ‘fetal-maternal HLA-(mis)matches’ are not known at the time of proposed use of the model, thereby rendering its inclusion impractical. We would like to clarify that currently fetal-maternal HLA testing is conducted in a research setting only. The implication of prenatal HLA typing into clinical practice remains a topic of ongoing research. We have added to the discussion section that future studies should carefully consider the financial implications of HLA typing as this may impact the feasibility of large-scale implementation and clinical practice.

*The analysis should include a comparative evaluation of the traditional tests on the same population, otherwise no significant difference could be calculated, since, as the Authors themselves admit, no data is published on the predictive performance of the available screenings on donor oocyte population.

- Thank you for the valuable suggestion. We are currently assessing literature on prediction models for hypertensive complications in pregnancy. It is worth noticing that none of the existing prediction models have been specifically developed in a population that conceived after assisted reproductive techniques only. Considering that the risk of developing hypertensive complications is significantly higher in oocyte donation pregnancies compared to naturally conceived pregnancies, it is plausible that a prediction model developed in a cohort of oocyte donation pregnancies only would perform better in this population.

Conversely, a model developed for autologous pregnancies may perform better in that population, compared to an oocyte donation-cohort only. Therefore, at this stage, we believe it may not be necessary to conduct a comparative evaluation of traditional tests on the same population.

We have added to the methods section that in order to enhance generalizability and applicability of a final model, predictor selection should be well defined, standardized and reproducible and that we based selection of predictors on literature and clinical reasoning.

*Last but not least, I suggest the use of a respectful language in the Introduction section: some sentences like the second one, lines 66-67 and 74-75, could be amended avoiding old fashioned expressions like "produce viable oocytes for healthy offspring" (not acceptable in 2023) and "lesbian motherhood".

- We regret that we have used old fashioned language, we did not intend to do so. Although to our knowledge, the term ‘shared lesbian motherhood’ is a common term used to describe the reception of oocytes from the partner, we have replaced ‘lesbian motherhood’ by ‘reception of oocytes from partner (ROPA)’ and replaced ‘women unable to produce viable oocytes for healthy offspring...’ by ‘women facing challenges in generating healthy oocytes for successful pregnancy’.

VERSION 2 – REVIEW

REVIEWER	Simeone, Serena University Hospital Careggi
REVIEW RETURNED	09-Mar-2024

GENERAL COMMENTS	Very interesting field of investigation. this aspect of preeclampsia needs to be addressed. Nevertheless, the number of pregnant OD participants treated with acetilsalicylic acid, once identified as high risk on the basis of the acog o fmf algorithm, could significantly reduce the proportion of overt preeclampsia. This possible bias need to be fixed and statistically considered. As a second advice, I'd suggest to use the more recent definition of PIH/HDP from ISSHP updated in 2022.
---

VERSION 2 – AUTHOR RESPONSE

Reviewer 1 (dr. Serena Simeone)'s comments (09-04-2024):

*Very interesting field of investigation. this aspect of preeclampsia needs to be addressed.

Nevertheless, the number of pregnant OD participants treated with acetilsalicylic acid, once identified as high risk on the basis of the acog o fmf algorithm, could significantly reduce the proportion of overt preeclampsia. This possible bias need to be fixed and statistically considered.

> Thank you for your valuable comment. It is indeed important to assess the predictive value of 'use of acetylsalicylic acid'. As a result, we have included 'use of acetylsalicylic acid' as a candidate predictor for our model, as depicted in the study outcomes section (methods). However, it should be noted that in predictive research causal effects are not investigated.

*As a second advice, I'd suggest to use the more recent definition of PIH/HDP from ISSHP updated in 2022.

> We have updated the study outcomes- and the model development sections accordingly and we have updated the references list.